

# Mutation breeding of *Aspergillus niger* by atmospheric room temperature plasma to enhance phosphorus solubilization ability

Qiuju Peng[1,*], Yang Xiao[2,*], Su Zhang[1,3], Changwei Zhou[1], Ailin Xie[1], Zhu Li[1,4], Aijuan Tan[1], Lihong Zhou[1], Yudan Xie[1], Jinyi Zhao[1], Chenglin Wu[1], Lei Luo[1], Jie Huang[1], Tengxia He[1] and Ran Sun[1]

[1] Key Laboratory of Plant Resource Conservation and Germplasm Innovation in Mountainous Region (Ministry of Education), Collaborative Innovation Center for Mountain Ecology & Agro-Bioengineering, College of Life Sciences/Institute of Agro-Bioengineering, Guizhou University, Guiyang, Guizou Province, China
[2] Institution of Supervision and Inspection Product Quality of Guizhou Province, Guiyang, China
[3] Bureau of Agriculture and Rural Affairs, Xixiu District, Anshun, Guizou Province, China
[4] Guizhou Key Laboratory of Agricultural Biotechnology, Guiyang, China
* These authors contributed equally to this work.

## ABSTRACT

**Background:** Phosphorus (P) is abundant in soils, including organic and inorganic forms. Nevertheless, most of P compounds cannot be absorbed and used by plants. *Aspergillus niger* v. Tiegh is a strain that can efficiently degrade P compounds in soils.
**Methods:** In this study, *A. niger* xj strain was mutated using Atmospheric Room Temperature Plasma (ARTP) technology and the strains were screened by Mo-Sb Colorimetry with strong P-solubilizing abilities.
**Results:** Compared with the *A. niger* xj strain, setting the treatment time of mutagenesis to 120 s, four positive mutant strains marked as xj 90–32, xj120–12, xj120–31, and xj180–22 had higher P-solubilizing rates by 50.3%, 57.5%, 55.9%, and 61.4%, respectively. Among them, the xj120–12 is a highly efficient P solubilizing and growth-promoting strain with good application prospects. The growth characteristics such as plant height, root length, and dry and fresh biomass of peanut (*Arachis hypogaea* L.) increased by 33.5%, 43.8%, 43.4%, and 33.6%, respectively. Besides available P, the chlorophyll and soluble protein contents also vary degrees of increase in the P-solubilizing mutant strains.
**Conclusions:** The results showed that the ARTP mutagenesis technology can improve the P solubilization abilities of the *A. niger* mutant strains and make the biomass of peanut plants was enhanced of mutant strains.

# INTRODUCTION

Phosphorus (P) is the second essential macronutrient for plant growth and development. Although P only accounts for 0.2% of plant dry weight, low P availability can generate a strong limit in plant growth and crop yield (*Sharma et al., 2013*). The natural reserve of soluble P fractions is much lower than expected, which may just have a range of doses

Corresponding authors
Zhu Li, zhuliluck@163.com
Aijuan Tan, 772771868@qq.com

from 1.0 µg L$^{-1}$ to 1.0 mg L$^{-1}$ in soil solutions. This is formed due to mineral P compounds that are easily combined with aluminum, calcium, iron, and magnesium together the tight fixation of in a form of soil colloids (*Kumar, Behl & Narula, 2001*; *Brady & Weil, 2000*). Therefore, many crops are suffering the problem of low P deficiency.

Phosphate-solubilizing microorganisms (PSM) play a great role in reducing P insufficiency in soils because they can provide low-cost soluble P fractions in different production systems (*Khan, Zaidi & Ahmad, 2014*). PSM have been shown to enhance the solubility of inorganic P compounds and increase crop yields (*Singh & Reddy, 2011*). The process of converting insoluble P into the soluble forms by PSM involves a series of stages, such as acidification, chelation, and exchange reactions (*Narsian & Patel, 2000*). The secretion of organic acids accounts for the major mechanism to make P fractions soluble (*Wahid & Mehana, 2000*). Organic acids of plant or microbial origin can form organometallic complexes with Fe$^{2+}$ and Al$^{3+}$, increasing P availability by the dissolution of phosphate precipitates with these ions (*Pavinato & Rosolem, 2008*). Previous studies have demonstrated that several mineral-solubilizing fungi could solubilize inorganic minerals into available forms through the production of organic acids (*e.g.*, citric, fumaric, gluconic, malic, succinic, tartaric, and oxalic acids) (*Acevedo et al., 2014*; *Wang et al., 2018*; *Khuna et al., 2021*). Specifically, the secretion of oxalic acid by strains under P-deficient soil conditions was one of the most important P-solubilization mechanisms that plants have adapted to cope with P deficiency (*Song et al., 2008*; *Nascimento et al., 2021*). Filamentous fungus *A. niger* is an outstanding cell factory for organic acid production due to its high carbon conversion efficiency and tolerance for low pH (*Meyer, Wu & Ram, 2011*; *Yang, Lübeck & Lübeck, 2016*). *A. niger* is a species widely used in the food industry to produce various enzymes and metabolites, such as citric acid (*Behera, 2020*). A gross estimation indicated that 80% of citric acids were fermented by *A. niger* all over the world (*Adeoye, Lateef & Gueguim-Kana, 2015*). *A. niger* ATCC 1015 exhibited great potential for malic acid production (*Xu et al., 2019*). The *A. niger* WH-2 could biosynthesize L (+)-tartaric acid under acidic conditions (*Bao et al., 2020*). The organic acids can chelate calcium, aluminum ferrous, and magnesium, which can all further increase P availability (*Mulik et al., 2020*). The P-solubilization activity of microorganisms is related to organic acid production. However, the nature of the acid produced was also important, dependent on the carbon source supplied (*Vassileva, Vassilev & Azcon, 1998*; *Srividya, Soumya & Pooja, 2009*). Therefore, choosing the appropriate carbon resource or other conditions to affect the nature of acid produced is important. *A. niger* could also promote plant growth by enhancing P uptake. For example, a study revealed that *A. niger* could promote the growth and nutrient uptake in groundnut (*Arachis hypogaea* L.) (*Jitendra, Kan & Vaibhavi, 2011*). Besides, *A. niger* also promoted growth in soybean (*Glycine max* L.) (*Elazouni, 2008*). Although the P-solubilizing ability of *A. niger* is well known, reports on improving the P-solubilizing ability of strain and promoting plant growth by microbial mutation breeding technique formulation are limited.

Microbial mutation breeding is an important technique, which has been broadly employed for microbial improvement, especially in the fields of biotechnology, biomanufacturing, food fermentation, and environmental protection (*Brock, 1976*).

Mutagenesis tools, including chemical mutagens, physical or biological approaches, are used to increase mutation rate and accelerate the subsequent evolutionary process to obtain mutant strains with desirable phenotypes (*Huang et al., 2021*). Among them, Atmospheric Room Temperature Plasma (ARTP) mutagenesis technology caused more serious DNA damage and a higher mutation rate among all the mutagenesis methods (*Zhang et al., 2015*). ARTP is a novel type of atmospheric pressure non-equilibrium discharge plasma source, which relies on a uniform distribution of highly concentrated neutral active particles to alter microbial genetic traits and directly modify the molecular structure at a nucleotide level (*Zong et al., 2012*). It has wide applications and high efficiency of mutation diversity and offers good safety and ease-of-use (*Zhang et al., 2014*). ARTP mutagenic technology has been successful in obtaining strains with desirable characteristics among bacteria, fungi, and microalgae (*Zhang et al., 2019a*, *2019b*). *Zong et al. (2012)* used ARTP technology to generate mutations in *Streptomyces albulus*, and the spores concentration was $10^6$ CFU mL$^{-1}$, which the positive mutation rates can reach a level as high as up to 26.0%. *S. albulus* A-29 exhibited the highest ε-Poly-L-lysine productivity, which was four times higher than that of the wild strain in the same culture condition. *Song et al. (2017)* revealed that, compared with the original strain, the lipid productivity of *Chlorella pyrenoidosa* was nearly improved over 16% by ARTP. The above examples illustrate that ARTP, as a novel mutagenesis tool, has a good application prospect.

This study aims to screen highly effective P-solubilizing strains of *A. niger* by ARTP mutagenesis and the peanut biomass and physiological indexes of the positive mutant strains were determined and investigated. This research provides a fundamental basis for the further development of multifunctional fungal agent and their applications. In addition, it provides a reference model and a method for the mutation breeding of other microorganisms.

## MATERIALS AND METHODS

### *Aspergillus niger* strain and reagents

The original strain *A. niger* strain (xj) was isolated and identified by the Institute of Fungi Resource, Guizhou University, and stored at the China Center for Type Culture Collection (CCTCC NO. M206021) (*Li et al., 2007*). Chemical reagents of this experiment were purchased from Sinopharm Chemical Reagent Co., Ltd. (China).

### Culture media

Potato dextrose agar (PDA) medium (g/L): the boiled potato extract 200, glucose 20, agar 20.

Pikovskaya's medium (g/L): glucose 10, sodium chloride 0.3, potassium chloride 0.3, ammonium sulfate 0.5, magnesium sulfate heptahydrate 0.3, ferric sulfate heptahydrate 0.03, manganese sulfate 0.03 and tricalcium phosphate 5.

## Mutation of *Aspergillus niger* xj

The method of mutagenesis of strains referred to *Zhang et al. (2019)*. Helium was used as the working gas and the mutagenic treatment distance was 2 mm, the processing power was 120 W, the gas flow rate was 10 slm (standard liter per minute), and the treatment time was from 0 s to 300 s with intervals of 30 s. The calculation of lethality rate was carried out following the protocol of *Luo et al. (2018)*.

$$\text{lethality rate} = \frac{U - T}{U} \times 100\%$$

where U represents the number of unmutated colonies (0 s) and T represents the number of colonies after the mutagenesis.

## Screening of positive mutant strains

Sterilized pikovskaya's agar medium was poured into sterilized petri dish. After solidification of the medium, a pinpoint inoculation of the single mutagenic colony of *A. niger* strain was made on the plates aseptically and incubated for 7 days at 28 °C. The intensity of tricalcium phosphate solubilisation was measured as the diameter of the halozone formed around the colonies (*Maurya & Kumar, 2006*). The above method aims to primarily screened the mutagenesis strains with P-solubilization effect. The primary screening strains were selected with a better P dissolving ability for re-screening, and P content was determined by Mo-Sb Colorimetry (*Li & Zheng, 2020*). The genetic variability of the mutant strains induced by artificial mutagenesis was low and easy to repair or mutate. Therefore, the mutant strains were subcultured based on their genetic stability (*Tao et al., 2004*). In this study, the genetic stability of the mutant strain was evaluated by 10 passages. Values were shown as mean ± standard deviation (SD) from triplicately repeated experiments.

## Measurement of organic acids

In this study, seven kinds of organic acids (oxalic acid, malic acid, citric acid, tartaric acid, lactic acid, acetic acid, and succinic acid) were determined by a method of *Yang et al. (2018)*. The organic acid concentration as the independent variable $x$ and the peak area as the dependent variable $y$, were taken to draw the standard curve by the external standard method. The unknown organic acids in culture filtrate were determined by comparing the retention times (RT) and peaks areas of chromatograms with the standard organic acids (*Bakri, 2019*).

## Determination of indole-3-acetic acid (IAA) content

IAA concentration (0, 10, 20, 30, 40, and 50 µg mL$^{-1}$) was taken separately, and a volume of 4 mL Salkowski was added to the mixture to promote full reaction at 28 °C for 30 min. IAA concentration was measured by UV-VIS Spectrophotometer (BIOMATE 3S; Thermo Fisher, Waltham, MA, USA) at 530 nm. IAA concentration as the abscissa $x$ and the absorbance as the ordinate $y$ to draw the IAA standard curve. *A. niger* spore suspension ($10^8$ CFU mL$^{-1}$) was inoculated and incubated at 28 °C and centrifuged at 150 r/min for 5 days. After that, it was centrifuged again at 109 g (Allegra X-30R; Beckman

Coulter, Brea, CA, USA) for 10 min, mixed with liquid supernatant (1 mL) and Salkowski (4 mL) at 28 °C for 30 min. Three technical replicates were done in the experiment, the IAA content was calculated by comparing the IAA standard curve.

## Determination of phosphorus content

The determination of P content was carried out by the method of *Zhang (2008)*. However, the absorbance of the sample was measured by UV spectrophotometer at 700 nm. The standard curve of phosphorus was drawn with P content as the abscissa $x$ and absorption value as the ordinate $y$. The content of soluble P in the centrifugal supernatant was determined by Mo-Sb Colorimetry.

## Determination of phosphorus removal kinetics curve of strains

The spore suspension was inoculated into a liquid medium containing 5% calcium phosphate. The culture solution was inoculated at 28 °C and 150 r/min for 7 days. During the culturing, the samples were taken at regular intervals of 24 h to be centrifuged at 109 g for 5 min at 4 °C. The concentration of spore suspension was set to $1 \times 10^8$ CFU/mL for xj90–32, xj120–12, xj120–31, and xj180–22 mutated strains. The content of soluble P in the centrifugal supernatant was determined and the value of pH was determined by pH meter (PHS-3C; Dapu Instrument, Shanghai, Co., Ltd., Shanghai, China).

## Effect of different factors on the phosphorus-solubilizing ability of mutant strains

Studies have shown that the carbon source, nitrogen source, pH, and other factors of the culture medium would affect the P-solubilizing ability of P-solubilizing fungi (*Patel, Archana & Kumar, 2008*). Therefore, the mutagenic strains of *A. niger* need to investigate the influence of these factors on the ability of solubilizing P. These factors included (carbon source and concentration, nitrogen source and concentration, temperature, NaCl, initial pH). Effects of changes in carbon sources and concentration, nitrogen sources and concentration, temperature, salinity, initial pH on P-solubilizing abilities were evaluated using calcium phosphate medium. The type of carbon sources included glucose, sucrose, maltose, lactose, and starch, and concentration were 0.5%, 1%, 5%, 10% and 15%. Baseline experimental conditions were: concentration of *A. niger* spore suspension $1 \times 10^8$ CFU mL$^{-1}$, ammonium sulfate concentration 0.5 g L$^{-1}$, initial pH 7.5, NaCl 3 g L$^{-1}$, and incubated aerobically (150 rpm min$^{-1}$, 30 °C) for 5 days. Nitrogen sources were derived from beef extract, sodium nitrate, ammonium sulfate, potassium nitrate, and carbamide, and concentration were 0%, 0.05%, 0.5%, 1%, 5% and 10%, NaCl concentrations were regulated to 0%, 0.5%, 1.5%, 2%, 3%, 4%, 5%, 6%, 8%, and 10%, initial pH was adjusted to 3, 5, 7, 9, and 11, and temperatures were changed to 20, 25, 30, 35, 40, and 50 °C. The effects of all these factors on the P solubilization were separately determined. Other conditions are the same as for the above determination of P content. Values are shown by mean ± SD from triplicate experiments.

## Collection of peanut samples and measurement of growth characteristics

Peanut seeds were firstly disinfected with hydrogen peroxide, rinsed with sterile water, and subjected to 28 °C for 24 h to accelerate the germination. Subsequently, seeds were placed in a test pot containing 300 g of soil, which was collected from the campus of Guizhou University with the geographical coordinates of 106 66 E′ and 26 44 N′. The larger particles and debris were filtered, which were mixed with fine sand in a 1:2 (V:V) ratio. Besides, a certain amount of insoluble calcium phosphate was added to each pot of soil at the ratio of 1:1,000 (W:W). The *A. niger* positive mutant strains ($10^6$ CFU $g^{-1}$) were inoculated with the peanut that was grown for 3 days, and the blank controls were prepared (inoculating the same amount of sterile water for CK control). When peanuts were growing up 30 days and was collected. Then, sterile water was used to clean the soil at the root of the peanut, and the filter paper was used to suck water. The growth indicators (plant height, root length, and dry and fresh biomass), chlorophyll content, and protein content were measured. the plant height and root length were measured by a ruler. Fresh biomass was measured with an analytical balance (BS110S; Sartorius Tianping, Co., Ltd., Beijing, China). Besides, the peanut was put into a sterilization carton and sterilized at 105 °C for 20 min and dried at 80 °C to constant weight. The dry biomass of the peanut was measured by analytical balance. Values shown are mean ± SD from triplicate experiments.

## Determination of physiological indexes (chlorophyll and total protein content)

Around 100 g of peanut leaves were collected, washed, and dried. They were next cut into slices in 1-mm width and placed in a 10 mL test tube. Leaves were fully soaked in 2 mL of dimethyl sulfoxide (DMSO) at 65 °C until they turned white or transparent. The extracted liquid was cooled to room temperature, and the extracted liquid was mixed fully with 80% acetone (8 mL) (*Qiu et al., 2016*). Chlorophyll concentration was determined using UV-VIS Spectrophotometer at 663.6 nm, and 646.6 nm and the calculation formula referred to *Porra, Thompson & Kriedemann (1989)*.

The standard protein curve of bovine serum protein was made by a method of *Bradford (1976)*. Around 0.5 g of fresh peanut leaves was taken, washed, and wipe-dried. The dried leaves were placed in a mortar and about 5 mL of phosphate buffer saline (PBS) was added to the mortar. Leaves were grounded to extract the protein, which was thoroughly mixed with Coomassie brilliant blue G-250 (0.1 mg $mL^{-1}$) and stand for 5 min. The absorbance value was measured at 595 nm by UV-VIS Spectrophotometer and the protein content in the sample was calculated. The formula is as follows:

$$\text{The protein content of sample (mg/g)} = \frac{C \times V_T V_1}{FW \times 1000}$$

where C represents the protein content calculated by the standard protein curve, unit (μg); $V_T$ represents the total volume of the extract, unit (mL); $V_1$ represents the added sample

amount when determination, unit (mL); FW represents the fresh weight of the sample, unit (g).

## Effect of the *Aspergillus niger* spore suspension on available phosphorus content in the soil

Soil samples were air-dried and filtered using a 1 mm sieve. Around 2 g of dried samples were transferred to a 100 mL conical flask filled with 20 mL of 0.50 mol $L^{-1}$ sodium bicarbonate solution. Then, samples were put at 28 °C and 150 r/min for 30 min. After, the samples were filtered and centrifuged at 109 g for 5 min. One mL of supernatant was used to determine the available P content by Mo-Sb Colorimetry. The test was performed in triplicate.

## Statistical analysis

One-way analysis of variance (ANOVA) was performed for data analysis using IBM SPSS (ver. 23.0; SPSS Inc., Armonk, NY, USA). The differences among the groups were evaluated by Duncan's test, and $p$-value < 0.05 was considered significant.

# RESULTS

## Investigation of the ARTP mutagenesis time

The lethality rate of the strain increased with time. The positive mutation rate showed a trend of increase first and then a decrease. When the treatment time reached 120 s, the lethality rate was 37%, and the positive mutation rate was 25.5%. The treatment time was 240 s, the lethality rate was 85%, whereas the positive mutation rate was only 6.38%. Therefore, the treatment time with the highest positive mutation rate was considered as the best condition for the mutagenesis of the xj strain, and the treatment time was 120 s (Fig. 1).

## Screening of mutagenic strains

Certain strains among a total of 670 mutant xj strains obtained by ARTP mutagenesis were efficient in solubilizing the inorganic phosphorus (calcium phosphate) in the medium. The 141 and 24 mutant strains were acquired by preliminary and secondary screening, respectively. The standard curve of P content was prepared and analyzed by linear regression, resulting in the following correlation model: $y = 0.0103x + 0.0069$, $r = 0.994$, which showed a good linear relationship and met the sample testing requirements.
The analysis results revealed that the xj90–32, xj120–12, xj120–31, and xj180–22 strains did not show differences in P-solubilizing abilities within 10 generations, indicating that the four strains were good genetic stabilities and the P-solubilizing abilities increased by 50.3%, 57.5%, 55.9%, and 61.4%, respectively (Table 1).

## Determination of organic acid contents

HPLC was used to analyze the organic acids in the medium containing calcium phosphate and revealed the P-solubilization mechanism of the xj strain. In this experiment, the standard curves of seven organic acids were prepared and analyzed by linear regression, resulting in the following coefficients: oxalic acid ($y = 0.0987x + 0.2843$, $r = 0.9944$), malic

 

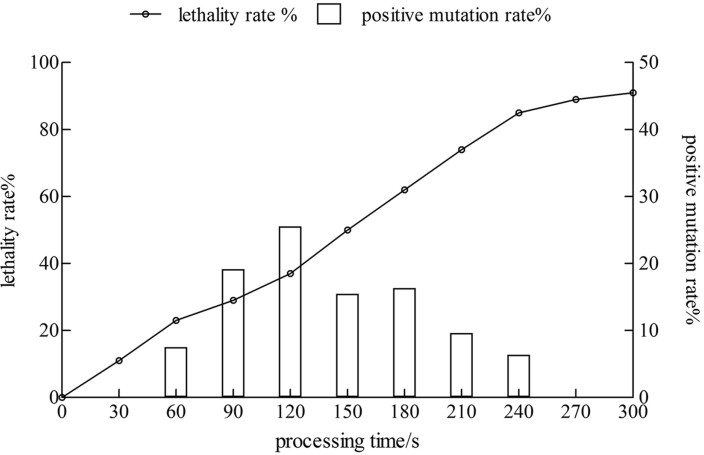

**Figure 1 The lethality rate and positive mutation rate of *Aspergillus niger* strain were determined by atmospheric toom temperature plasm mutagenesis.**

**Table 1 The results of genetic stability testing of mutant strains of *Aspergillus niger*.**

| Fungi strains | Phosphate-solubilizing quantity (µg/mL) (generation) | | | | | | | | | | p-values | |
|---|---|---|---|---|---|---|---|---|---|---|---|---|
| | One | Two | Three | Four | Five | Six | Seven | Eight | Nine | Ten | Mean value | |
| xj | 508.3 ± 16.3 | 512.9 ± 9.4 | 518.3 ± 24.2 | 520.2 ± 31.4 | 502.4 ± 19.8 | 498.3 ± 30.1 | 490.1 ± 18.3 | 491.5 ± 22.6 | 480.7 ± 20.5 | 497.1 ± 18.8 | 502.0 | >0.05 |
| xj90–32 | 776.2 ± 25.8 | 775.1 ± 17.3 | 770.0 ± 23.2 | 761.2 ± 3.2 | 736.6 ± 20.7 | 750.5 ± 38.5 | 746.0 ± 8.2 | 745.7 ± 19.3 | 753.9 ± 29.6 | 748.6 ± 38.2 | 754.4 | >0.05 |
| xj120–12 | 792.1 ± 19.8 | 781.6 ± 24.5 | 800.5 ± 25.1 | 787.2 ± 25.01 | 786.4 ± 28.7 | 777.8 ± 10.3 | 776.1 ± 17.2 | 795.5 ± 33.2 | 808.1 ± 27.8 | 801.4 ± 20.8 | 790.6 | >0.05 |
| xj120–31 | 776.8 ± 12.4 | 788.1 ± 28.8 | 774.8 ± 30.3 | 779.9 ± 19.5 | 776.5 ± 30.2 | 781.3 ± 21.9 | 782.3 ± 34.35 | 781.15 ± 21.3 | 789.8 ± 32.2 | 791.9 ± 17.1 | 782.8 | >0.05 |
| xj180–22 | 810.1 ± 29.2 | 828.0 ± 61 | 823.8 ± 69.6 | 819.9 ± 1.62 | 832.8 ± 6.5 | 822.1 ± 24.1 | 818.1 ± 24.1 | 816.8 ± 23.4 | 812.5 ± 24.1 | 807.1 ± 37.4 | 819.1 | >0.05 |

**Note:**
$P < 0.05$ means a significant difference among means using Duncan's test.

acid ($y = 0.0228x + 0.118$, $r = 0.9995$), citric acid ($y = 0.0101x + 0.2628$, $r = 0.9984$), tartaric acid ($y = 0.0146x + 0.7278$, $r = 0.9946$), lactic acid ($y = 0.0057x + 0.2372$, $r = 0.9987$), acetic acid ($y = 0.0055x + 0.6848$, $r = 0.9951$) and succinic acid ($y = 0.0055x + 0.4042$, $r = 0.9983$). All above-mentioned curves revealed close linear relationships, which met the requirements of samples testing. We observed differences in the kinds and contents of organic acids between the xj strain and the four positive mutant strains (Table 2).

To be more specific, the xj strain and xj90–32, xj120–12, xj120–31, and xj180–22 strains secreted oxalic acid with contents reaching the peaks of 1,363.7 mg L$^{-1}$, 1,426.6 mg L$^{-1}$, 1,402.4 mg L$^{-1}$, 1,329.4 mg L$^{-1}$, and 1,394.2 mg L$^{-1}$, respectively and also secreted citric acid, the contents reached the maximum of 140.2 mg L$^{-1}$, 151.4 mg L$^{-1}$, 156.3 mg L$^{-1}$, 145.6 mg L$^{-1}$, and 172.9 mg L$^{-1}$, respectively. In addition, the xj120–12 strain could secret succinic acid and the highest content was 543.5 mg L$^{-1}$.

## Determination of IAA content

The standard curve of IAA was prepared and analyzed by linear correlation, resulting in the following coefficient: $y = 0.0138x + 0.0138$, and $r = 0.9972$. The curve displayed a good linear relationship, which met the requirements of sample testing. As shown in

**Table 2 Contents of organic acids secreted by the *Aspergillus niger* strain within 5 days.**

Organic acid content (mg· L$^{-1}$)

| Fungi strains | Time | Oxalicacid | Malic acid | Citric acid | Tartaric acid | Lactic acid | Acetic acid | Succinic acid |
|---|---|---|---|---|---|---|---|---|
| xj | 1d | 730.5 | – | 110.1 | – | – | – | – |
| | 2d | 942.3 | – | 112.2 | – | – | – | – |
| | 3d | 1,363.7 | – | 140.2 | – | – | – | – |
| | 4d | 1,239.2 | – | 134.2 | – | – | – | – |
| | 5d | 1,216.3 | – | 135.8 | – | – | – | – |
| xj90–32 | 1d | 843.5 | – | 142.0 | – | – | – | – |
| | 2d | 1,002.4 | – | 148.4 | – | – | – | – |
| | 3d | 1,424.6 | – | 151.4 | – | – | – | – |
| | 4d | 1,339.2 | – | 141.1 | – | – | – | – |
| | 5d | 1,304.1 | – | 138.5 | – | – | – | – |
| xj120–12 | 1d | 830.5 | – | 143.9 | – | – | – | – |
| | 2d | 1,209.2 | – | 148.2 | – | – | – | 543.5 |
| | 3d | 1,402.4 | – | 156.3 | – | – | – | 201.7 |
| | 4d | 1,299.3 | – | 151.2 | – | – | – | – |
| | 5d | 1,205.3 | – | 148.4 | – | – | – | – |
| xj120–31 | 1d | 893.2 | – | 121.5 | – | – | – | – |
| | 2d | 1,022.3 | – | 134.2 | – | – | – | – |
| | 3d | 1,329.4 | – | 145.6 | – | – | – | – |
| | 4d | 1,293.8 | – | 142.8 | – | – | – | – |
| | 5d | 1,221.6 | – | 138.7 | – | – | – | – |
| xj180–22 | 1d | 784.3 | – | 161.4 | – | – | – | – |
| | 2d | 973.2 | – | 165.4 | – | – | – | – |
| | 3d | 1,394.2 | – | 172.9 | – | – | – | – |
| | 4d | 1,302.5 | – | 170.4 | – | – | – | – |
| | 5d | 1,259.5 | – | 162.8 | – | – | – | – |

Fig. 2 the xj and four mutated strains could produce IAA in a medium containing 0.5 g L$^{-1}$ tryptophan. The positive mutant strain xj120–12 exhibited a significant difference from xj strains ($P < 0.05$). The highest IAA content was 10.5%, which was higher than that in xj strain. However, strains of xj90–32, xj120–31, and xj180–22 were not significantly different from xj strains ($P > 0.05$).

## Determination of kinetic curve of phosphorus solution

According to Fig. 3A, all strains started releasing soluble P into the culture medium on day first. On the fifth day, the soluble P content of xj, xj90–32, xj120–12, xj120–31, and xj180–22 strains reached the maximum values of 554.3 μg mL$^{-1}$, 744.4 μg mL$^{-1}$, 787.6 μg mL$^{-1}$, 769.9 μg mL$^{-1}$, and 810.9 μg mL$^{-1}$, respectively. Afterward, the P content began decreasing at 6 and 7 days affecting by time. Simultaneously, the pH value of five strains sharply decreased during the first 2 days and remained at a lower pH value

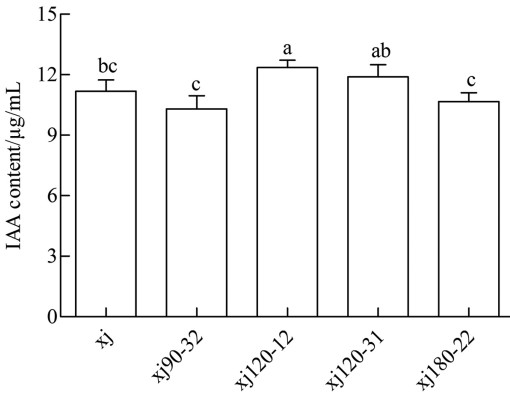

**Figure 2 The concentrations of IAA produced by *Aspergillus niger* xj and mutant strains.** The vertical bars indicate standard deviation of biological replicate and different lowercase letters indicate significant differences ($P < 0.05$).

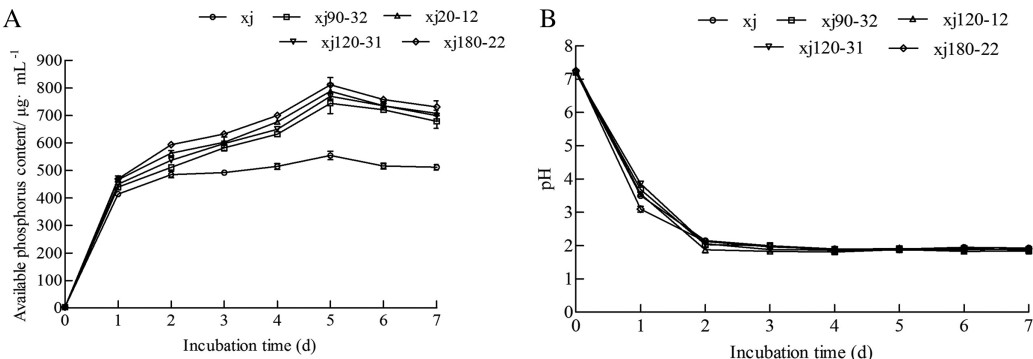

**Figure 3 The change of soluble P (A) and pH (B) during incubation by 7 days for *Aspergillus niger* xj and mutant strains.** Data are expressed as the average of three experiments. Error bars represent mean ± SD.

from 2 to 7 days (Fig. 3B). During the dissolution of calcium phosphate, the soluble P content of five strains presented a relatively negative relationship with pH.

## Effect of carbon sources on the phosphorus-solubilization ability of strain

The effects of different carbon sources (monosaccharides, disaccharides, and polysaccharides) on the ability of *A. niger* to dissolve calcium P were investigated. Effects of different carbon sources on the P content were significantly different in order, as follows: glucose > sucrose > maltose > starch > lactose (Fig. 4A). When glucose was used as the sole carbohydrate component, P content was the highest ($P < 0.05$). Compared with the xj strain, the P-solubilizing rates of four positive mutant strains marked as xj90–32, xj120–12, xj120–31, xj180–22 increased by 41.2%, 49.2%, 45.8%, and 52.4%, respectively. In addition, glucose concentration had a significant effect on the P-solubilizing ability of the strains. In the range of 5~15%, the P-solubilizing increased with the increase of

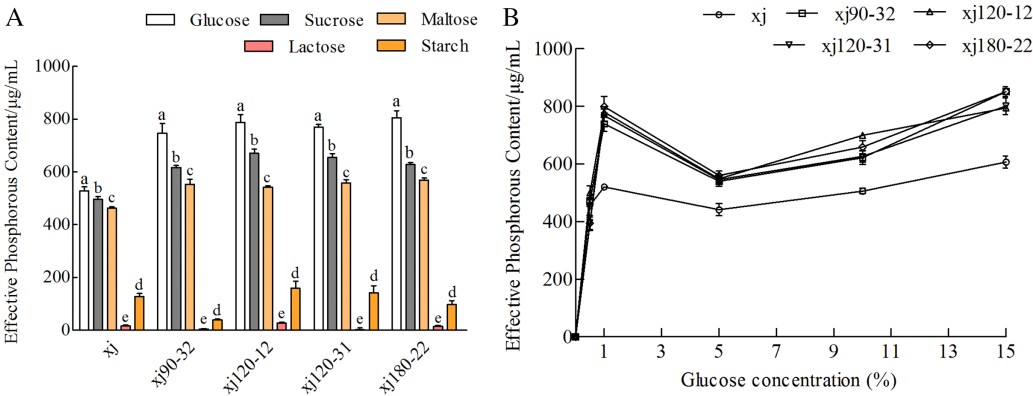

**Figure 4 The effects of carbon sources (A) and glucose concentration (B) on the P-dissolving capability of *Aspergillus niger* xj and mutant strains.** Different lowercase letters indicate significant differences, and the lowercase letters used for the significance comparisons apply only to comparisons within the same strain ($P < 0.05$).

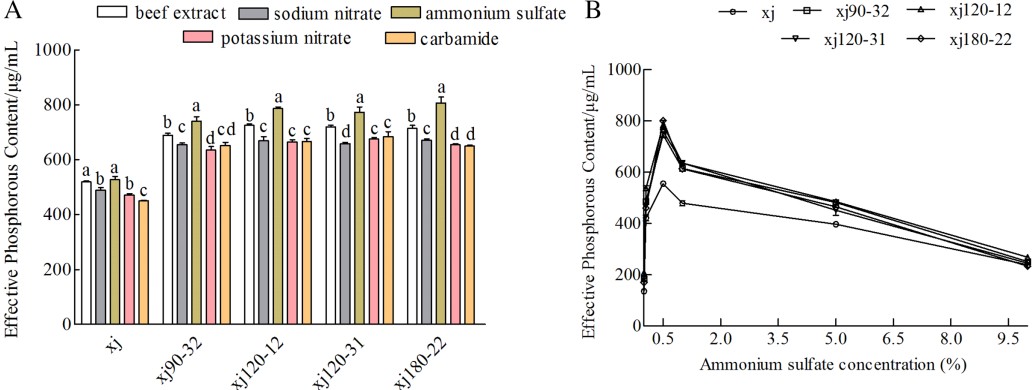

**Figure 5 The effect of nitrogen sources (A) and ammonium sulfate concentration (B) on the P-dissolving capability of *Aspergillus niger* xj and mutant strains.** Different lowercase letters mean significant differences, and the lowercase letters used for the significance comparisons apply only to comparisons within the same strain ($P < 0.05$).

glucose concentration. Compared to the xj strain, the P-solubilizing rates of four positive mutant strains improved 40.1%, 30.9%, 31.2%, and 40.1%, respectively (Fig. 4B).

## Effect of nitrogen sources on the phosphorus-solubilization ability of strain

Nitrogen sources had different influences on the P removal ability of the strains. When ammonium sulfate was used as the nitrogen component, the P content was the highest ($P < 0.05$). In contrast to the xj strain, the xj90–32, xj120–12, xj120–31, and xj180–22 strains had higher P-solubilizing rates boosted by 40.5%, 49.1%, 46.6%, and 53.1%, respectively (Fig. 5A). Besides, when ammonium sulfate concentration was 0.5%, the P-solubilizing quantities of the mutant strains enhanced 34.1%, 41%, 36.6%, and 62.2%, respectively (Fig. 5B).

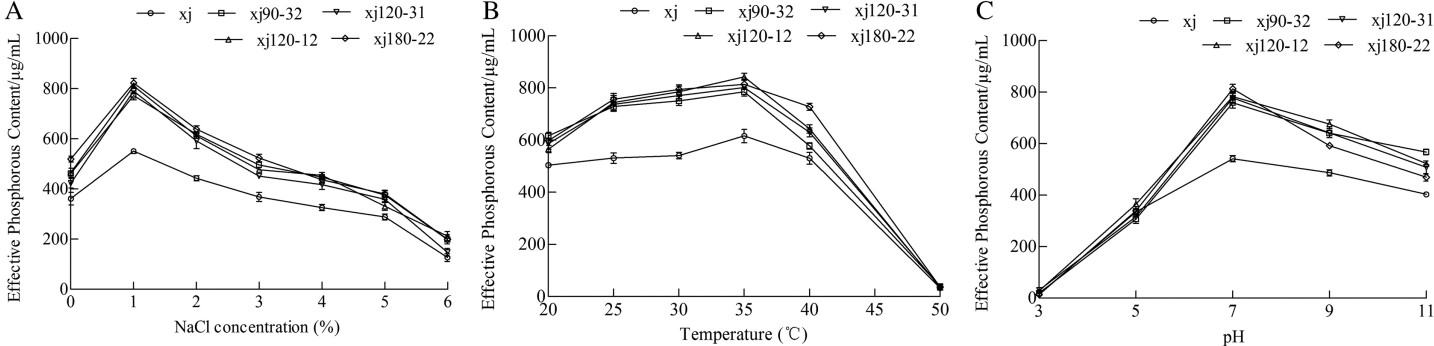

**Figure 6 The effect of NaCl concentration (A), temperature (B), and pH value (C) on the P-dissolving capacity of *Aspergillus niger* xj and mutant strains.** Data are expressed as the average of three experiments. Error bars represent mean ± SD.

## Effect of NaCl, temperature and pH on the phosphorus-solubilization ability of strain

The effects of different NaCl concentrations on the ability of strains to dissolve calcium P were determined. The P-solubilizing abilities of strains were the highest at lower NaCl concentration 1% (Fig. 6A). Compared to xj strain, the P-solubilizing quantities of strains (xj90–32, xj120–12, xj120–31, and xj180–22) increased by 40%, 47.1%, 43.4%, and 49.2%, respectively. The effect of temperature ranging from 20 to 50 °C on the ability of *A. niger* to hydrolyze calcium phosphate was determined. The P-solubilizing ability was the highest at the temperature 35 °C. Compared to the xj strain, P-solubilization quantities of strains (xj90–32, xj120–12, xj120–31, and xj180–22) increased by 27.4%, 36.9%, 30.2%, and 32.2%, respectively (Fig. 6B). To assess the pH adaptability of *A. niger* strains during P solubilization, the effect of pH (ranging from 3 to 11) on the ability of five strains to dissolve calcium P was determined. The P-solubilizing abilities of strains had the highest at pH 7.0. Compared to the xj strain, the P-solubilization quantities of strains (xj90–32, xj120–12, xj120–31, and xj180–22) enhanced 40%, 44.3%, 43.5%, and 50%, respectively (Fig. 6C).

## Effect of the *Aspergillus niger* spore suspension on the biomass of peanut plant

As it is shown in Fig. 7, the positive mutant strains showed different effects on the biomass of *peanut*. The xj120–12, xj90–32, and xj180–22 strains significantly promoted the growth of height of peanut ($P < 0.05$), which the biomass increased 68.7%, 55%, 47%, respectively higher than the CK and enhanced 33.5%, 22%, 16%, respectively, more than the xj strain (Fig 7A). As it is shown in Fig. 7B, the xj120–12, xj120–31, xj180–22, and xj90–32 strains obviously improved the growth of root length of peanut ($P < 0.05$). Their biomass improved 132.7%, 103%, 100%, 90.9%, respectively, higher than the CK. They enhanced by 43.8%, 26%, 24%, and 18%, respectively, over than the xj strain. As it is shown in Fig. 7C, compared to the CK, the xj120–12, xj180–22, and xj120–31 strains evidently increased the fresh biomass of 67.7%, 61%, and 40%, respectively. Simultaneously they improved 43.4%, 38%, and 20%, respectively higher than the xj strain. In addition,

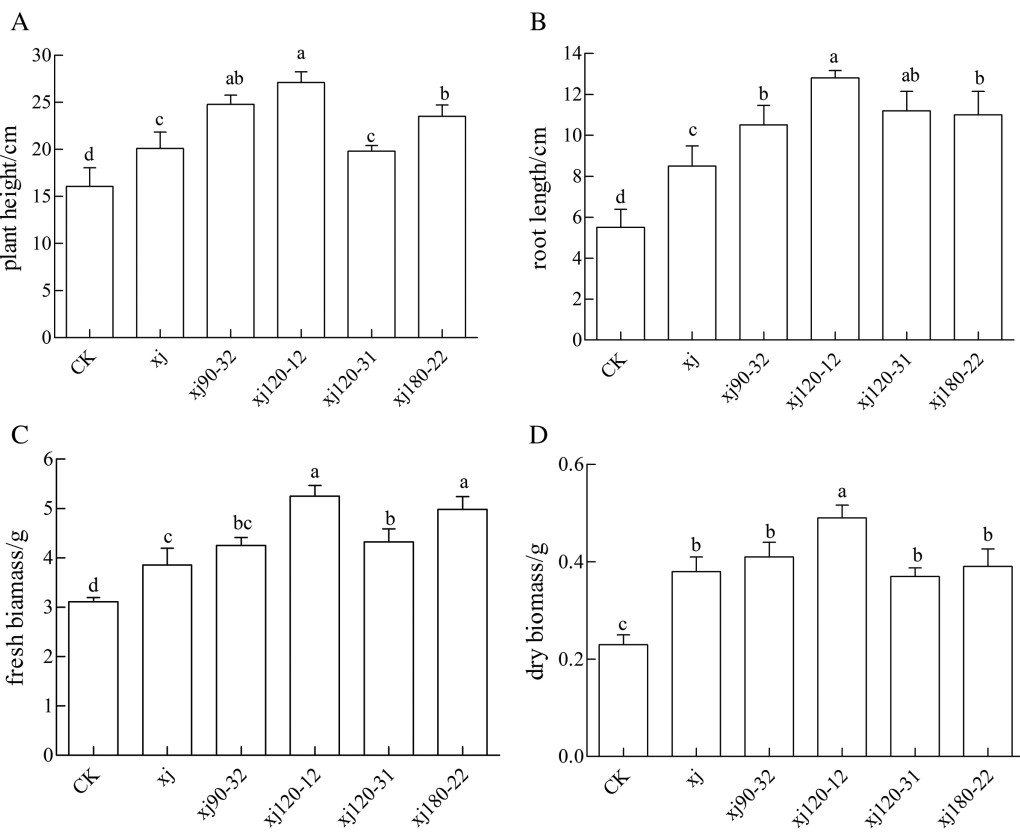

**Figure 7 The effects of *Aspergillus niger* xj and mutant strains on plant height (A), root length (B), fresh biomass (C) and dry biomass (D) of peanut grown in pot experiment with soil sand mixture by 30 days.** CK represented blank group. Different lowercase letters indicate significant differences ($P < 0.05$).

compared to the CK, the xj120–12 strain markedly boosted the dry biomass of 117.2%. It improved 33.6% than the xj strain (Fig 7D).

## Effect of mutant positive strain on physiological indexes

The strains influenced the physiological indexes of peanuts, including chlorophyll and soluble protein content (Fig 8). The xj180–22 and xj120–12 strains improved the chlorophyll content compared with the CK ($P < 0.05$) by 49.5% and 27.9%, respectively. The enhancements were 36% and 16%, respectively, higher than the xj strain (Fig. 8A). In addition, the positive mutant strains increased soluble protein content than the CK ($P < 0.05$). The soluble protein content of xj180–22, xj120–31, xj120–12, and xj90–32 strains increased 12%, 10.7%, 9.2%, and 6%, respectively, than the CK. However, compared with the xj strain, the soluble protein contents were not significantly different in all the positive mutant strains ($P > 0.05$) (Fig. 8B).

## Effect of mutant strain on the available phosphorus content in the soil

The available P content in soil for the CK was 6.88 mg kg$^{-1}$, which was considered low. The inoculation with the xj spore suspension increased the P content to 12.85 mg kg$^{-1}$. The xj120–12, xj120–31, xj90–32 and xj180-22 increased by 169.4%, 127.6%, 121.3% and

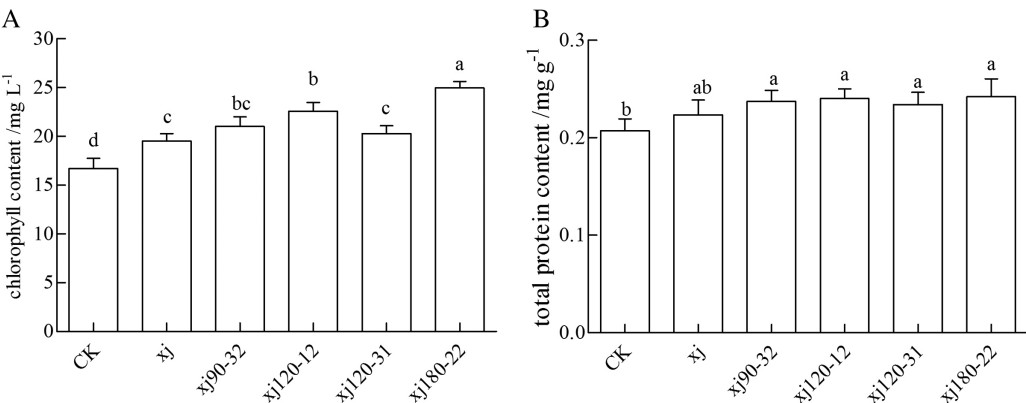

**Figure 8 The effects of *Aspergillus niger* xj and mutant strains on physiological indexes (chlorophyll content and soluble protein content).** CK represented blank group. Different lowercase letters indicate significant differences ($P < 0.05$).

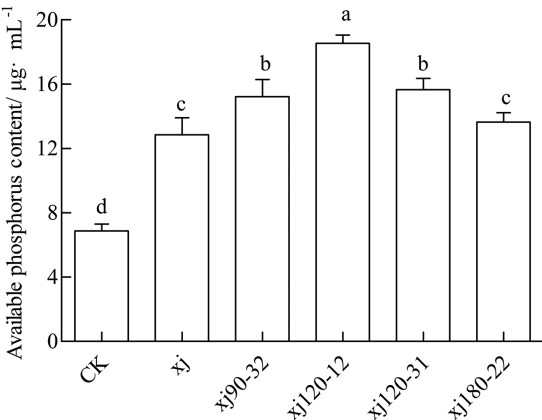

**Figure 9 Effects of *Aspergillus niger* xj and mutant strains on the content of plant available phosphorus in the soil after cultivation of peanut in pot experiment with soil sand mixture by 30 days.** CK represented blank group. Different lowercase letters indicate significant differences ($P < 0.05$).

98.3%, respectively, higher than CK ($P < 0.05$). In addition, the xj120–12, xj120–31 and xj90–32 boosted 44.3%, 21.9%, and 18.5%, respectively higher than the xj strain ($P < 0.05$) (Fig. 9). In general, inoculation with the mutant strains xj120–12 could improve the P content in soils and promote peanut growth.

## DISCUSSION

With the establishment of ARTP mutation and using primary screening, re-screening and subcultured 10 passages, we eventually obtained four representative mutants: xj90–32, xj120–12, xj120–31, and xj180–22. The positive mutation rate was 25.5%, when the optimal treatment time was 120 s. The good genetically stable of four strains were obtained, and the P-solubilization abilities increased more than 50%. Due to the occurrence of positive mutations is related to the mutagenic parameters and characteristics

of the strain (*Zhang et al., 2019*). The positive mutation rate of strains were the highest in the study when the fatality rate was 37%.

With the decrease in the pH value, P content increased by contrast. This indicated that pH could be related to the P-solubilization ability of the mutant strain. Several reports have revealed that pH decreased due to the release of organic acids (*Nath, Sharma & Barooah, 2012*; *Padamavathi, 2015*; *Whitelaw, 2000*). Oxalic acid and critic acid were detected in four positive mutant strains and the xj strain. In addition, succinic acids were also secreted by xj120–12. The nature of organic acid produced is a specific characteristic of each fungal isolate but may vary under different cultivation conditions. Arguably, the mutagenesis treatment affected the kinds of organic acid, resulting in differences in metabolic capabilities and affecting the P-solubilization ability of mutant strains.

P-solubilization is a complex phenomenon that depends on many factors such as the culture's nutritional, physiological, and growth conditions (*Isbelia et al., 1999*). In this experiment, the kinetics curve showed an optimal culture time of 5 days, which was taken as the optimal culture time for the influencing factors. Glucose was used as a carbon source, and the P-solubilization abilities were obviously increased. Glucose is a monosaccharide with a simple structure and facilitates the acid production for the strain (*Wang et al., 2015*). Using ammonium sulfate as nitrogen sources also caused high P-solubilization efficiency. Similar results have been reported in previous studies that nitrogen in the ammonium form is necessary for P solubilization in organic acid production through $NH^+_4/H^+$ exchange mechanisms (*Omar, 1997*). The P-solubilizing abilities of the four mutant strains were the highest at 1% NaCl concentration, afterward, decreased with the NaCl concentration increase. These results were similar to the of *Srividya, Soumya & Pooja (2009)*. The reason is that too much chloride ions may chelate or neutralize proton ions or acid produced in the media then, decrease the P-solubilizing ability. Except for carbon source, nitrogen source type and concentration and NaCl concentration, the formation process, and growth of *A. niger* mycelium will be affected by temperature and pH value. In this study, the P-solubilizing ability started decreasing with temperature and pH value over 35 °C and 7, suggesting that microbial metabolism and biological activity affected the P-solubilizing ability of the strain. In summary, choosing the composition of the medium and conditions would increase the strains' P-solubilization ability; thus, improving the utilization rate of P in the soil (*Liu et al., 2015*).

The four positive mutant strains could increase in different degrees on height, root length, and dry and fresh biomass of peanut. The reason may be that the strains secreted IAA. According to reports, IAA is an endogenous plant growth hormone, promoting seed germination and thus plant growth (*Wu et al., 2014*). The xj180–22 and xj120–12 strains could improve the chlorophyll content. The cause is that chlorophyll is the most important pigment in plant photosynthesis, and its content is positively correlated with leaf color, which can roughly reflect the nutritional status of plants and affect the photosynthetic rate of plants (*Zhang, Zhang & Li, 2008*). The soluble protein as a kind of nutrient substance could directly affect the respiration of plants and catalyze all kinds of chemical reactions (*Berry, 1982*). In this study, the four mutant strains improved the

soluble protein content compared with the CK. Besides, the xj120–12 strain improved the P content in the soil.

## CONCLUSIONS

To sum up, the xj120–12 can be used as the dominant fungi with the potential of P solubilization and growth promotion and provided the basis for the production of P solubilization fungal fertilizer.

### Funding

This work was supported by grants from the Science and Technology Plan Project of Guizhou Province [Support of Guizhou science and technology cooperation [2021]193], National Natural Science Foundation of China [31660533; 31460486], The Open Fund Project of Guizhou Province Domestic First-Class Discipline Construction in Biology (GNYL[2017]009 FX3KT20), National First-Class Undergraduate Professional Construction Project in biological Sciences of Guizhou University, and Modern industrial technology system of traditional Chinese medicine in Guizhou Province. The funders had no role in study design, data collection and analysis, decision to publish, or preparation of the manuscript.

### Grant Disclosures

The following grant information was disclosed by the authors:
Science and Technology Plan Project of Guizhou Province: [2021]193.
National Natural Science Foundation of China: 31660533 and 31460486.
Guizhou Province Domestic First-Class Discipline Construction in Biology: GNYL[2017] 009 FX3KT20.
Guizhou University Biological Science National First-Class Undergraduate Major Construction Project.
Modern Industrial Technology System of Traditional Chinese Medicine in Guizhou Province.

### Competing Interests

We have a patent number for the *Aspergillus niger* xj strain (CN1847388).

### Author Contributions

- Qiuju Peng conceived and designed the experiments, performed the experiments, analyzed the data, prepared figures and/or tables, authored or reviewed drafts of the paper, and approved the final draft.
- Yang Xiao conceived and designed the experiments, performed the experiments, analyzed the data, prepared figures and/or tables, authored or reviewed drafts of the paper, and approved the final draft.
- Su Zhang conceived and designed the experiments, performed the experiments, prepared figures and/or tables, and approved the final draft.

- Changwei Zhou conceived and designed the experiments, prepared figures and/or tables, and approved the final draft.
- Ailin Xie conceived and designed the experiments, authored or reviewed drafts of the paper, and approved the final draft.
- Zhu Li analyzed the data, authored or reviewed drafts of the paper, and approved the final draft.
- Aijuan Tan analyzed the data, authored or reviewed drafts of the paper, and approved the final draft.
- Lihong Zhou analyzed the data, prepared figures and/or tables, and approved the final draft.
- Yudan Xie analyzed the data, prepared figures and/or tables, and approved the final draft.
- Jinyi Zhao analyzed the data, prepared figures and/or tables, and approved the final draft.
- Chenglin Wu performed the experiments, authored or reviewed drafts of the paper, and approved the final draft.
- Lei Luo analyzed the data, prepared figures and/or tables, and approved the final draft.
- Jie Huang analyzed the data, prepared figures and/or tables, and approved the final draft.
- Tengxia He performed the experiments, authored or reviewed drafts of the paper, and approved the final draft.
- Ran Sun performed the experiments, authored or reviewed drafts of the paper, and approved the final draft.

## Patent Disclosures

The following patent dependencies were disclosed by the authors:

*Aspergillus niger* xj strain patent number: CN1847388.

## Data Availability

The raw data are available in the Supplemental File.

## Supplemental Information

Supplemental information for this article can be found online at http://dx.doi.org/10.7717/peerj.13076#supplemental-information.

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
