# Peer review of "Mutation breeding of Aspergillus niger by atmospheric room temperature plasma to enhance phosphorus solubilization ability"

_PeerJ, doi:10.7717/peerj.13076_

## Round 0.1 · original submission · Minor Revisions

Our reviewers have a few suggestions for improvement. I also have one of my own: in figures 4 and 5 the letters showing which comparisons are statistically significant have been wrongly chosen. That must be corrected: it is highly advisable to not repeat the same letter with a different meaning in the same graph (e.g. one should not use the letter "a" for the glucose bar in strain xj and the same letter for the glucose bar in the xj90-32 strain unless one intends to state that the xj glucose bar labelled "a" is not significantly different from the xj90-32 glucose bar labelled "a").

Reviewer 1 ·

Basic reporting

This manuscript is readable and technically correct except for the following errors which should be corrected.

L332-333 there is no the data of xj180–22. And “xj190–32”? Is it xj90–32?

L343-344 What is the difference compared with others? Please point it out.

Experimental design

This manuscript is suitable for publication for its research questions are well defined and meaningful. However the following questions should be answered before publication.

L77-79 Silva U et al. (PLoS ONE, 2014. doi.org/10.1371/journal.pone.0110246) have showed UV mutagenesis can also improve the P-solubilizing of Aspergillus niger. Please explain why you choose ARTP not UV.

L110 Please write down the ARTP parameters, for instance temperature, helium flow rate and irradiation distance. It is very important for this study.

L119-120 What are the standards for the preliminary screening? The author did not describe it clearly. The size of halo zone of dissolving phosphate? Please describe it in detail.

L226-227 Generally, we think the lethality rate of 50~60% is better and we can get more positive strains. However, the results showed that the positive mutation rate was the highest when the lethal rate was 37%. Please explain the reason for this phenomenon.

L319-320 P-solubilizing abilities of xj180–22 is better than xj120–12 under different influencing factors. Why is xj120–12 better for the biomass of peanut plant than xj180–22?

Validity of the findings

It is meaningful for this manuscript to find a fungi xj120–12 to be used as a dominant fungi with the potential of P solubilization and growth promotion and provide the basis for the production of P solubilization fungi fertilizer.

Reviewer 2 ·

Basic reporting

1.This article text is quite professional English language used throughout. Please check Line 388 : fungi fertilizer should be fungal fertilizer?
Structure conforms to PeerJ standards, discipline norm, or improved for clarity.
Figures are relevant, high quality, well labelled & described.
2. Your introduction needs more detail. I suggest that you improve at lines 49 by adding ref Kuna et al. 2021, Frontiers in Microbiology about the Aspergillus section Nigre for mineral solubilizing activities to provide more information in general properties.

3.Reference This part is very weak need to improve much more for consistency of writing pattern according to Journal rule.
1. Scientific name need to be in italic ex L396, L472, L486, L488, L490,L517, L535
2. Journal name should be in full and not in italic. Check how to write journal name.
3. Title of article : check constancy of writing. Only the beginning of sentence that first letter is capital the other word is in small letter

Experimental design

Original primary research within Scope of the journal.
Research question well defined, relevant& meaningful. It is stated how the research fills an identified knowledge gap. Rigorous investigation performed to ahigh technical & ethical standard.
Methods described with sufficient detail & information to replicate.
Question 1. Author wrote in methodology L 183 add Ca3(PO4)2 1:1000 this is by weight or by volume ratio?
Question 2: Line 184 author use inoculum at 106 CFU why not use 1x108 CFUg-1 as state before?

Validity of the findings

Impact and novelty are provided.
All underlying data have been provided; they are robust, statistically sound, & controlled.
Conclusions are well stated, linked to original research question & limited tosupporting results

Additional comments

The manuscript is clearly written in professional, unambiguous language. If there is a weakness, it is in the writing of References pattern (as I have noted above) which should be improved upon before Acceptance

---

## Round 0.2 · Minor Revisions

I am generally satisfied with your responses to our reviewers' queries, but I am afraid that the letter labeling in Figs. 4 and 5 is still deficient: for example, all bars in the glucose section of fig 4A are labeled "a" in spite of the control strain being obviously different from the others. A given letter should mean "this bar is significantly different from all other bars in the graph that bear a different letter" AND "this bar is NOT significantly different from any other bar that bears the same letter". You are instead probably using letters a-e to compare growth in different substrates for a given strain and then when you are comparing growth in different substrates for another strain you are again using the letters a-e. This is, unfortunately, ambiguous and must be corrected.

---

## Round 0.3 · Minor Revisions

I have seen the changes you made to figures 4 and 5 and I fear that they are still not at all clear.

Please see the picture I attach, to better understand what my issue is. I also suggest that instead of using double letters (Aa,Ab,...) you use lowercase a-z letters and repeat a letter every time that two bars are not significantly different. This means all letters from a-y will be used only if all 25*24/2 comparisons are significantly different, but if (for example) the third bar is not significantly different from the 1st or the 2nd (while the 2nd bar is significantly different from the 1st) the 1st bar will be labelled "a", the 2nd will be labelled "b" and the 3rd will be labelled "ab" (it takes the "a" letter because it is indistinguishable from the "a" bar and the "b" because it is indistinguishable from the "b" bar)

I am eager to accept your manuscript but I must ensure that readers have no trouble understanding which comparisons are significantly different or not.

---

## Round 0.4 · accepted · Accept

All the issues have been solved to my satisfaction. I am glad to accept your paper for publication.